# Bacterial OTU deubiquitinases regulate substrate ubiquitination upon Legionella infection

Donghyuk Shin[1,2,3,4], Anshu Bhattacharya[1,2], Yi-Lin Cheng[1,2], Marta Campos Alonso[2], Ahmad Reza Mehdipour[3], Gerbrand J van der Heden van Noort[5], Huib Ovaa[5†], Gerhard Hummer[3,6], Ivan Dikic[1,2,3]*

[1]Institute of Biochemistry II, Faculty of Medicine, Goethe University Frankfurt, Frankfurt, Germany; [2]Buchmann Institute for Molecular Life Sciences, Goethe University Frankfurt, Frankfurt, Germany; [3]Max Planck Institute of Biophysics, Frankfurt, Germany; [4]Department of Nano-Bioengineering, Incheon National University, Incheon, Republic of Korea; [5]Oncode Institute and Department of Cell and Chemical Biology, Leiden University Medical Centre, Leiden, Netherlands; [6]Institute of Biophysics, Goethe University Frankfurt, Frankfurt, Germany

**Abstract** *Legionella pneumophila* causes a severe pneumonia known as Legionnaires' disease. During the infection, Legionella injects more than 300 effector proteins into host cells. Among them are enzymes involved in altering the host-ubiquitination system. Here, we identified two **L**egionella**OT**U (ovarian tumor)-like deubiquitinases (LOT-DUBs; LotB [Lpg1621/Ceg23] and LotC [Lpg2529]). The crystal structure of the LotC catalytic core (LotC$_{14-310}$) was determined at 2.4 Å. Unlike the classical OTU-family, the LOT-family shows an extended helical lobe between the Cys-loop and the variable loop, which defines them as a unique class of OTU-DUBs. LotB has an additional ubiquitin-binding site (S1'), which enables the specific cleavage of Lys63-linked polyubiquitin chains. By contrast, LotC only contains the S1 site and cleaves different species of ubiquitin chains. MS analysis of LotB and LotC identified different categories of host-interacting proteins and substrates. Together, our results provide new structural insights into bacterial OTU-DUBs and indicate distinct roles in host–pathogen interactions.

*For correspondence:
dikic@biochem2.uni-frankfurt.de

† Deceased

## Introduction

Ubiquitination, a well-studied post-translational modification, regulates various cellular events (*Yau and Rape, 2016*). A representative example of the ubiquitin-mediated cellular process is the ubiquitin-proteasome system, where misfolded proteins get ubiquitinated, degraded by the proteasome, and, finally, recycled (*Dikic, 2017*). For larger cellular waste, such as cellular components (endoplasmic reticulum [ER], mitochondria, etc.), protein aggregates, or intracellular bacteria, ubiquitination works together with the autophagy machinery, which includes the sequestration of ubiquitinated components and their transfer into the lysosome for degradation (*Pohl and Dikic, 2019*). To maintain homeostasis in the cell, ubiquitination events are tightly regulated by a reverse process called deubiquitination, where ubiquitin molecules are specifically cleaved from the target substrates and subsequently recycled by deubiquitinating enzymes (deubiquitinases [DUBs]) (*Clague et al., 2019*).

To date, about 100 different DUBs have been identified in human. They are categorized into seven different classes based on their structure and mechanism of action, and these include USP, JAMM (MPN), OTU, MJD (Josepin), UCH, and the recently discovered MINDY and ZUFSP

(*Abdul Rehman et al., 2016*; *Clague et al., 2019*; *Haahr et al., 2018*; *Hermanns et al., 2018*; *Hewings et al., 2018*; *Kwasna et al., 2018*). Six of them belong to the cysteine protease family (USP, OTU, MJD [Josepin], UCH, MINDY, and ZUFSP), while JAMM (MPN) belongs to the zinc-containing metalloproteases. Among them, the OTU-family is distinguished from other DUBs, as they exhibit linkage specificity (*Mevissen et al., 2016*; *Mevissen et al., 2013*; *Mevissen and Komander, 2017*). For example, Cezanne specifically cleaves Lys11-linked polyubiquitin chains (*Bremm et al., 2010*) and OTUB1 preferentially cleaves Lys48-linked chains (*Edelmann et al., 2009*; *Wang et al., 2009*), while OTULIN exclusively cleaves M1-linked (linear) chains (*Keusekotten et al., 2013*). Extensive biochemical and structural studies have provided general mechanisms of the diverse linkage specificity within the structurally similar OTU-family. In general, the selectivity is achieved by the specific orientation of the S1 site, which accepts proximal ubiquitin and the S1' site that binds primed ubiquitin of ubiquitin chains. Besides, the presence or the absence of additional ubiquitin-binding domains (UBDs), sequence variations on ubiquitinated substrates, or S2-binding site that binds to the third ubiquitin within the chains can also affect the specificity of OTU-family (*Mevissen et al., 2013*).

Considering the importance of ubiquitin-mediated cellular pathways, it is not surprising that pathogens are armed with various weapons to hijack the host-ubiquitination system. For instance, *Salmonella typhimurium* encodes HECT type E3 ligase SopA (*Diao et al., 2008*; *Fiskin et al., 2017*; *Lin et al., 2012*) and *Legionella pneumophila* contains LubX and LegU1, which are similar to U-box- and F-box-containing E3 ligases, respectively (*Ensminger and Isberg, 2010*; *Kubori et al., 2008*; *Quaile et al., 2015*). In addition, bacterial pathogens possess atypical ubiquitin ligases that do not belong to any of the known E3 ligases, such as IpaH family (Shigella) or SidC/SdCA (Legionella) (*Hsu et al., 2014*; *Wasilko et al., 2018*). More recently, the SidE family (SdeA, SdeB, SdeC, and SidC) of Legionella has been shown to mediate unconventional phosphoribosyl (PR) serine ubiquitination mechanism, which is also tightly regulated by the meta-effector SidJ or PR-ubiquitin-specific DUBs (DupA and DupB) (*Bhogaraju et al., 2019*; *Bhogaraju et al., 2016*; *Black et al., 2019*; *Kalayil et al., 2018*; *Qiu et al., 2016*; *Shin et al., 2020*). Pathogenic bacteria encode not only ubiquitin ligases but also DUBs (*Hermanns and Hofmann, 2019*). The most studied bacterial DUBs are CE-clan proteases, based on the MEROPS database classification, which cleave either ubiquitin or ubiquitin-like modifiers (SUMO1 or NEDD8) (*Pruneda et al., 2016*; *Rawlings et al., 2018*). In addition to CE-clan DUBs, bacteria and viruses encode OTU-like DUBs. Several structures from viral-OTUs revealed that they have a unique structure compared to those of known OTU family members (*Akutsu et al., 2011*; *Capodagli et al., 2013*; *James et al., 2011*; *Lombardi et al., 2013*; *van Kasteren et al., 2013*). OTUs from nairovirus Crimean-Congo hemorrhagic fever virus (CCHFV) and Dugbe virus (DUGV) have an additional β-hairpin in their S1-binding site. While viral OTUs have been studied extensively, only three bacterial OTU-like DUBs have been identified to date. ChlaOTU from *Chlamydophila pneumoniae* contains an OTU-domain that cleaves both K48- and K63-linked polyubiquitin chains (*Furtado et al., 2013*). LotA (Lpg2248; Lem21), Legionella <u>OT</u>U (LOT)-like DUB, contains two OTU-like domains with two catalytic Cys residues (C13 and C303), both of which are required for cleaving ubiquitin chains from an LCV (Legionella-containing vacuole) (*Kubori et al., 2018*). Interestingly, LotA showed K6-linkage preference that is solely dependent on its first OTU domain (Cys13). Recently, another OTU-like DUB from Legionella (Lpg1621, Ceg23) has been identified as K63 chain-specific OTU-DUB (*Ma et al., 2020*).

Despite these findings, little is known about the structure and molecular details of bacterial OTU-like DUBs. Here, we describe two novel OTU-like DUBs in Legionella – LotB (Lpg1621; Ceg23) and LotC (Lpg2529). Structural analysis of the LOT-DUBs provides insights into how bacterial OTU-DUBs are distinguished from the known OTU members. Furthermore, we also identified the specific host-substrates or interacting proteins of LotB and LotC by mass-spectrometry (MS) analysis using catalytically inactive variants. Collectively, our findings provide valuable structural insights into bacterial DUBs and their roles in host–pathogen interactions.

# Results

## Identification of two novel OTU-like DUBs from Legionella effector proteins

To identify putative DUBs amongst the Legionella effector proteins, we analyzed effector proteins from *L. pneumophila* (Lpg genes). Based on the type IV Icm/Dot complex secretion signal (>2.0), 305 effector proteins were selected (*Burstein et al., 2016*). Using pairwise sequence-structure comparison based on hidden Markov models (HMMs, HHpred suite) (*Zimmermann et al., 2018*), we revealed four previously uncharacterized proteins as putative DUBs. These proteins all contain catalytic domains of known DUBs (*Figure 1a*). Lpg1621(Ceg23) and Lpg2529 are found as members of the OTU-family, whereas Lpg2411 and Lpg2907 belong to the UCH and CE-clan, respectively (*Figure 1b*, *Table 1*). An in vitro di-ubiquitin cleavage assay with di-Ub panel (eight different linkage-specific $Ub_2$ chains [*El Oualid et al., 2010*]) showed that the OTU-like DUBs (Lpg1621 and Lpg2529) are capable of cleaving $Ub_2$ chains with different specificity, while other candidates

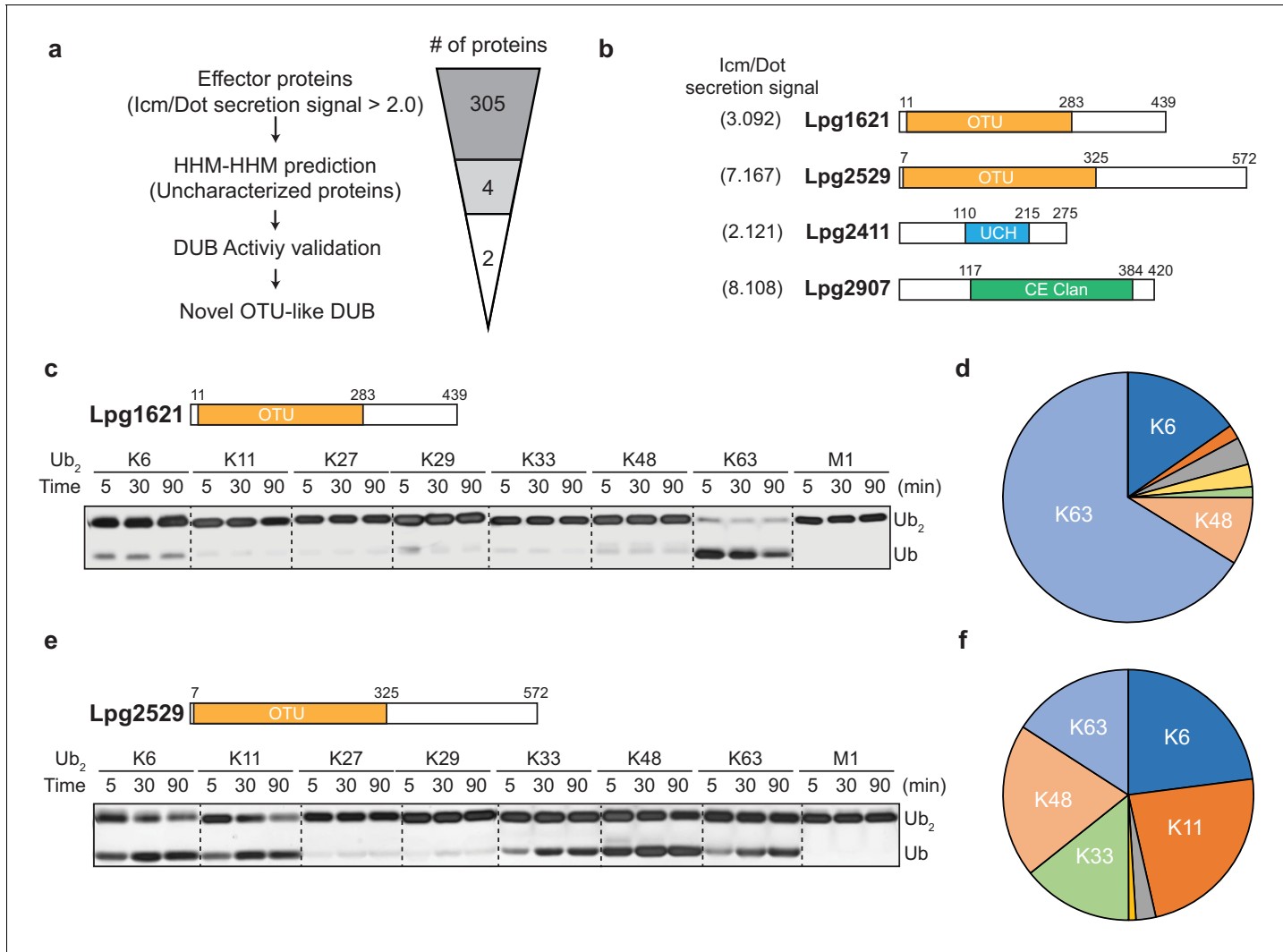

**Figure 1.** Identification of novel deubiquitinases (DUBs) in *Legionella pneumophila*. (a) Graphical illustration of identification of novel DUBs from *L. pneumophila* effector proteins. (b) Predicted DUB domain of four putative Legionella DUBs. (c, e) Time-course di-ubiquitin panel cleavage assay with Lpg1621 (LotB) and Lpg2529 (LotC). (d, f) Linkage specificity diagram of Lpg1621 (LotB) and Lpg2529 (LotC). The percentage of cleaved ubiquitin species at 90 min was plotted.

The online version of this article includes the following figure supplement(s) for figure 1:

**Figure supplement 1.** Ubiquitin cleavage assay with putative deubiquitinases (DUBs) from Legionella.

**Table 1.** TOP five candidates for putative deubiquitinases (DUBs) from Legionella effector proteins.

| Legionella proteins | | Target proteins | | | | |
|---|---|---|---|---|---|---|
| Name | Aligned region | Name | Aligned region | Probability (%) | Identities (%) | PDB ID_Chain |
| Lpg1621 | 195–274 | Viral OTU (CC hemorrhagic fever virus) | 69–157 | 92.59 | 16 | 3PHU_B |
| | 195–274 | Human OTUD2 | 63–140 | 92.52 | 17 | 4BOQ_A |
| | 195–279 | Human OTUD3 | 59–142 | 92.40 | 13 | 4BOU_A |
| | 193–278 | Human OTUD5 | 100–183 | 91.18 | 21 | 3PFY_A |
| | 192–279 | Viral OTU (Farallon virus) | 88–183 | 91.08 | 16 | 6D × 5_B |
| Lpg2529 | 1–310 | Viral OTU (Erve virus) | 17–157 | 96.24 | 18 | 5JZE_A |
| | 7–310 | Viral OTU (Dera Ghazi Khan orthonairovirus) | 25–156 | 96.15 | 18 | 6D × 2_B |
| | 20–310 | Human Otubain1 | 50–234 | 96.02 | 13 | 2ZFY_A |
| | 20–310 | Human Otubain2 | 50–233 | 95.84 | 13 | 4FJV_C |
| | 7–310 | Viral OTU (Taggert virus) | 23–156 | 95.71 | 13 | 6D × 3_D |
| Lpg2411 | 110–216 | Yeast UCH8 | 152–259 | 37.94 | 11 | 3MHS_A |
| | 183–272 | EntA-im (Enterococcus faecium) | 7–89 | 37.06 | 15 | 2BL8_B |
| | 33–94 | Uncharacterized protein (Corynebacterium diphtheriae) | 7–72 | 36.65 | 21 | 3KDQ_D |
| | 120–212 | PG0816 (Porphyromonas gingivalis) | 53–139 | 35.31 | 16 | 2APL_A |
| | 104–114 | PSII reaction center protein K (Cyanidium caldarium) | 2–12 | 32.96 | 36 | 4YUU_X2 |
| Lpg2907 | 117–384 | AvrA (Salmonella typhimurium) | 59–299 | 100 | 11 | 6BE0_A |
| | 158–398 | PopP2 (Arabidopsis thaliana) | 99–339 | 99.93 | 13 | 5W3X_C |
| | 115–390 | HopZ1a (Pseudomonas syringae) | 54–342 | 99.88 | 10 | 5KLP_C |
| | 118–275 | XopD (Xanthomonas campestris) | 1–148 | 95.53 | 11 | 2OIX_A |
| | 95–276 | Human SENP1 | 2–177 | 95.21 | 16 | 2G4D_A |

Values are obtained from the HHpred server (MPI Bioinformatics Toolkit).

(Lpg2411 and Lpg2907) did not show catalytic activity (*Figure 1—figure supplement 1*). The OTU family DUBs have been shown to have linkage specificity against certain polyubiquitin chains (*Mevissen et al., 2016*; *Mevissen et al., 2013*). To address whether Lpg1621 and Lpg2529 follow this fundamental rule, we performed a time-course in vitro DUB assay with di-Ub panel (*Figure 1c–f*). Consistent with the recent evidence, Lpg1621 exclusively processed the K63-linked $Ub_2$ (*Ma et al., 2020*), while Lpg2529 showed activity against K6-, K11-, K33-, K48-, and K63- linked $Ub_2$. Based on the sequence homology and catalytic activity, we have now renamed the Lpg1621 and Lpg2529 as LOT-like DUBs (LotB and LotC, respectively).

## Biochemical properties of LotB and LotC

The OTU-family belongs to the cysteine protease family, which requires the presence of a catalytic triad for their activity (*Mevissen et al., 2013*). Based on the sequence analysis, we identified the conserved catalytic triad for both LotB (D27, C29, and H270) and LotC (D17, C24, and H304). Mutations on either cysteine or histidine completely abolished the catalytic activity of both DUBs, suggesting that both LotB and LotC follow the general catalytic mechanism of the OTU-family (*Figure 2a and c*). Next, we sought to find whether LotB and LotC require additional ubiquitin-binding sites (S1' or S2). To elicit this information, we used two different types of ubiquitin activity-based probes (ABPs). The propargyl-di-ubiquitin-ABP (Prg-ABP) contains a highly reactive propargyl group at the C-terminus of ubiquitin chains, which can target S1 and S2 pocket (third-generation probes)

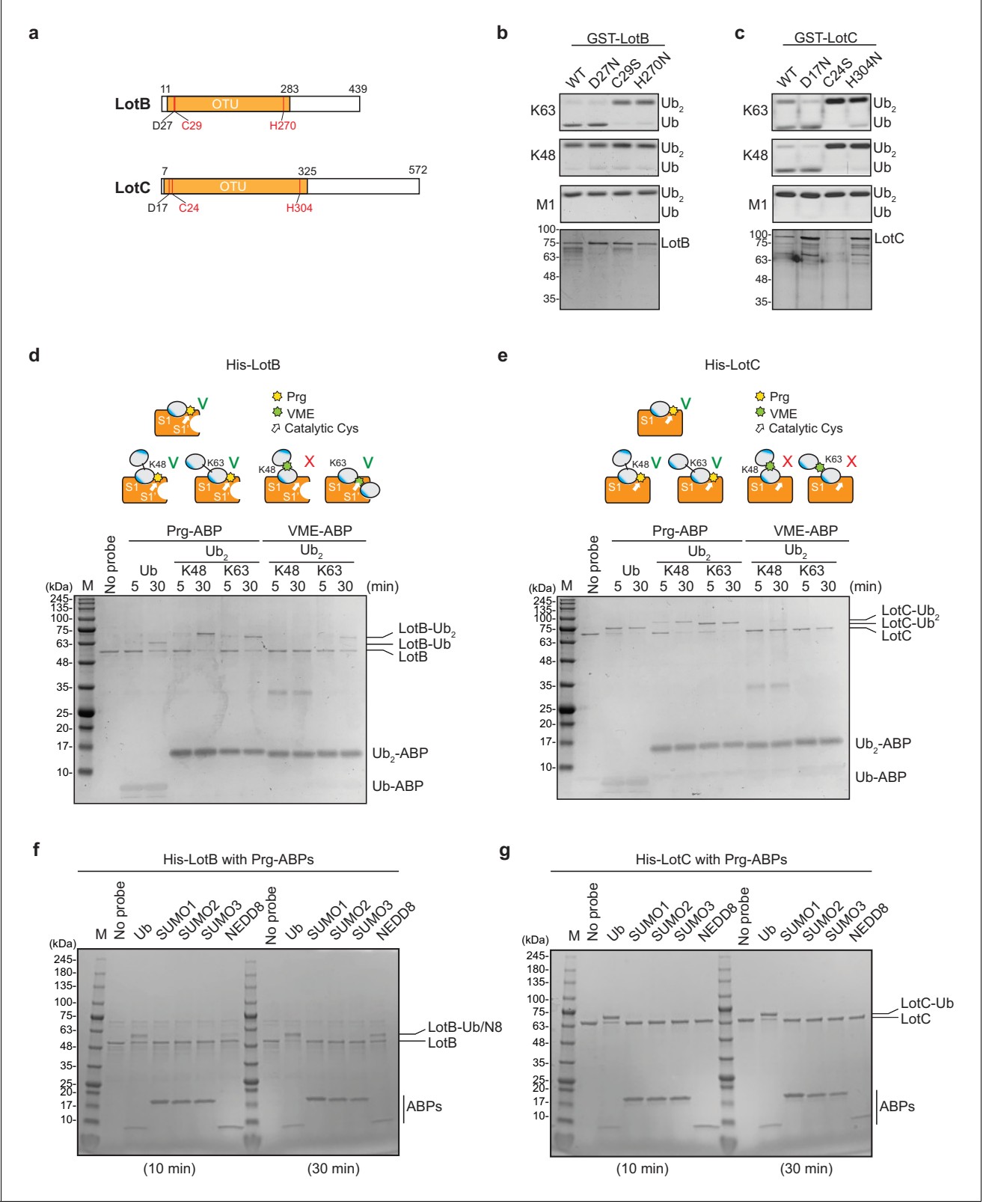

**Figure 2.** Biochemical properties of LotB and LotC. (**a**) Predicted catalytic residues on LotB and LotC. (**b, c**) Di-Ub cleavage activity assay with wild-type and catalytic mutants of LotB and LotC. (**d, e**) Activity-based probes (ABPs) test on LotB and LotC. Propargyl-Ub-ABP (Prg-ABP) and vinylmethylester-ubiquitin-ABP (VME-ABP) were incubated as indicated time-points with LotB and LotC and analyzed on SDS-PAGE with coomassie staining. (**f, g**) Propargyl ubiquitin or ubiquitin-like modifiers reactivity test on LotB and LotC. Prg-ABPs are incubated with LotB and LotC with indicated time points.

*Figure 2 continued on next page*

*Figure 2 continued*

The online version of this article includes the following figure supplement(s) for figure 2:

**Figure supplement 1.** Biochemical properties of LotB and LotC.

and form a covalent bond with the catalytic cysteine (*Ekkebus et al., 2013*; *Flierman et al., 2016*; *Sommer et al., 2013*). The vinyl methyl ester-ubiquitin-ABP (VME-ABP) contains VME, which replaces the isopeptide bond between two ubiquitin moieties in chains, which can detect S1 and S1' pocket (second-generation probe), and also forms a covalent bond with the catalytic cysteine (*Borodovsky et al., 2002*; *Mulder et al., 2014*). Both LotB and LotC showed clear shifts with all Prg-ABPs (mono-, K48-, and K63- linked), with different reactivity. LotB only partially shifted after 30 min, as evident by the amount of unreacted species, while LotC rapidly reacted with Prg-ABP and was completely conjugated after 30 min (*Figure 2d and e*). These results suggest that both LotB and LotC have a primary ubiquitin-binding S1 site, where the propargyl group can be located in close proximity to the catalytic cysteine. In contrast with Prg-ABP, only LotB reacted with K63-Ub$_2$-VME-ABP, which is consistent with the di-Ub panel assay (*Figure 1c–f*), where LotB showed specificity toward the K63-linkage. The VME-ABP results suggest that there is an additional S1' ubiquitin-binding site on LotB, which helps to properly locate the K63-linked-VME group on the catalytic site proximal to the catalytic cysteine. LotC lacks this S1' site, causing the VME group between two ubiquitin moieties to be unable to reach the catalytic cysteine. Next, we asked whether both LotB and LotC can interact with other ubiquitin-like modifiers (SUMO1/2/3, NEDD8) (*Figure 2f and g*, *Figure 2—figure supplement 1a and b*). Interestingly, LotB showed modification with both NEDD8-Prg and ubiquitin-Prg after a 30 min reaction, while LotC was modified only with ubiquitin. This suggests that LotB binds to both ubiquitin and NEDD8 through the conserved Ile44-mediated hydrophobic interactions, while LotC interacts with ubiquitin through specific residues present only in ubiquitin (*Figure 2—figure supplement 1c*).

## Structural analysis of LOT-like DUBs

Linkage specificity of the OTU family relies on one of the following mechanisms: (1) additional UBDs, (2) ubiquitinated sequences in the substrates, or (3) defined S1' or S2 ubiquitin-binding sites (*Mevissen et al., 2013*). To determine the minimal OTU domain for biochemical and structural studies, we designed several constructs and tested their activity against the di-Ub panel (*Figure 3a and b*). While LotC retained its activity with the predicted OTU domain (7–310), LotB lost its activity after deletion of 50 amino acids (300–350) located at the C-terminus, beyond the predicted OTU domain (11–283). Based on the LotB structure (PDB:6KS5, *Ma et al., 2020*), we assumed that this additional helical region might be required for the another ubiquitin-binding site (S1') to accept the distal ubiquitin moiety from K63 Ub$_2$ (*Figure 3c*). To understand the detailed mechanism of the linkage specificity of LotB and LotC at the molecular level, we determined the crystal structure of the catalytic domain of LotC (LotC$_{14-310}$) at 2.4 Å (*Figure 3d*, *Supplementary file 1*, PDB ID: 6YK8). A structural comparison of both LotB and LotC with other OTU-DUBs predicted by HHpred revealed that both Lot-DUBs have the unique structural features in the S1 ubiquitin-binding site (*Figure 3c and d* and *Table 1*). Whereas the overall fold of the catalytic core of LotB and LotC resembles that of other OTU-DUBs, both showed apparent differences in the helical arm region, which has been shown to serve as an S1-binding site and interact with ubiquitin (*Mevissen et al., 2013*). The structure and sequence alignment with other OTUs clearly showed that both LotB and LotC contain a relatively long insertion between the Cys-loop and the variable loop, compared to other OTU members (*Figure 3e*). The typical length of the helical lobe of the known OTUs is ranging from 50 to 60 amino acids (except the Otubain family, which contains 110–120 amino acids). In comparison, the helical lobes in LotB and LotC contain 183 and 210 amino acids, respectively. Based on this observation, we wondered whether LotA, another LOT-DUB (*Kubori et al., 2018*), also contains this extended insertion in the same region. Based on the catalytic cysteine and histidine residues of the two OTU domains on LotA (*Hermanns and Hofmann, 2019*), we analyzed the sequence and found that both OTU domains of LotA contain the extended insertion between the Cys loop and the variable loop (179 and 178 amino acids, respectively; *Figure 3e*). Together, our results identify Lot-

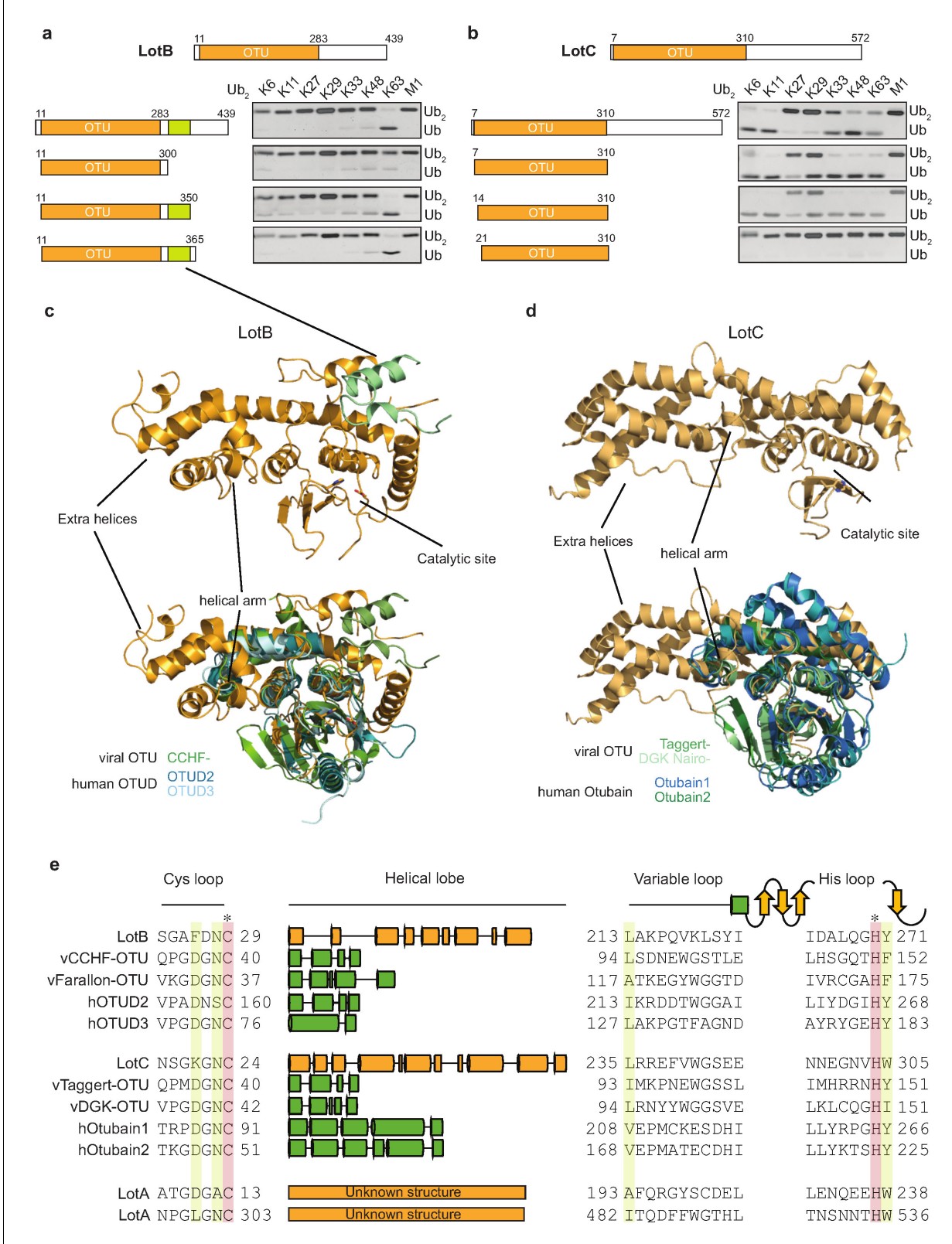

**Figure 3.** Structural comparison of Legionella OTU-deubiquitinases with other OTU-family. (**a, b**) Minimal domain boundaries of catalytically active LotB and LotC. Different constructs were cloned based on the predicted OTU-domains and their activity, and were tested with di-Ub panel. (**c, d**) Structural comparison of LotB and LotC with the closest homologues. CCHF- (PDB: 3PHU), OTUD2 (PDB: 4BOQ), OTUD3 (PDB: 4BOU), Taggert- (PDB: 6D × 3),

*Figure 3 continued on next page*

*Figure 3 continued*

DGK nairo- (PDB: 6D × 2), Otubain1 (PDB: 2ZFY), Otubain2 (PDB: 4FJV). (**e**) Sequence alignment of LotB and LotC with their closest homologues. Catalytic cysteine and histidine are highlighted in red and conserved residues are highlighted in yellow.

The online version of this article includes the following figure supplement(s) for figure 3:

**Figure supplement 1.** Sequence alignment of OTU deubiquitinase family.

DUBs as a novel class of the OTU-family with longer insertions in the helical lobe region (*Figure 3— figure supplement 1a*).

## A novel structural fold of S1 ubiquitin-binding sites on LOTs

Both LotB and LotC have extended helices, specifically near the S1 ubiquitin-binding site and we wondered how these regions interact with ubiquitin. To address this, we performed ubiquitin docking into both LotB and LotC, followed by molecular dynamics (MD) simulations for 600 ns (*Figure 4a–d*). The final models showed that ubiquitin indeed makes contacts with the additional helical regions of both LotB and LotC. In LotB, Phe143 and Met144 form hydrophobic interactions with ubiquitin (Phe45 and Ala46). In addition to these interactions, we found another hydrophobic patch in LotB (Ile238, Val247, Ala248, Ile264, and Ala266) to interact with ubiquitin (Leu71 and Leu73). For LotC, we identified several hydrophobic interactions of the extended helical region (Tyr119 and Tyr149) with ubiquitin (Ile44). During the simulation, the C-terminus of ubiquitin (Arg72 and Arg74) formed transient electrostatic interactions with LotC (Glu153 and Glu245). To validate the interactions observed in the simulations, we introduced several mutations to the binding interface of both LotB and LotC, and performed a ubiquitin-cleavage assay (*Figure 4e and f*). Consistent with a recent study (*Ma et al., 2020*), mutations of both F143 and M144 decreased the catalytic activity of LotB. Interestingly, mutations of the newly identified hydrophobic patch (I238 and A266) also reduced the catalytic activity. For LotC, the mutations in the hydrophobic patch (Y119 and Y149) affected its catalytic activity. Remarkably, a single mutation on E153 completely abolished the catalytic activity, which indicates that the electrostatic interactions are essential for the correct docking of the ubiquitin C-terminus into the catalytic pocket of LotC. This observation also explains the result observed in ubiquitin-like protein ABP assays with LotC (*Figure 2g*). The NEDD8, which has an alanine instead of Arg72 in ubiquitin, showed no modification toward LotC (*Figure 2— figure supplement 1c*). Together, our results reveal how the extra helical lobes of the Lot-DUBs interact with ubiquitin and how they differ within the LOT family.

## Proteomic studies of LotB and LotC

To gain better insights into the physiological functions of LotB and LotC, we decided to identify their interacting proteins or substrates. First, to enrich the interacting partners, catalytically inactive LotB or LotC were expressed in cells and immuno-precipitated from cell lysates. Ubiquitin (UBB) is strongly enriched with both catalytically inactive LotB and LotC (*Figure 5a and b*). MS analysis revealed that LotB mainly interacts with membrane protein complexes (COPB1, ATP5B, ATP5H, COX5A, and SEC61B). We also found the interaction of LotB with some ER-resident proteins (Calnexin [CANX], DDOST, STT3A). By contrast, most of the enriched proteins from the inactive LotC pull-down were non-membrane-bound organelle- and ribosome-related proteins (RPS8, RPLP2, RPS27, RPLP1, and RPL13) (*Figure 5a* and b). To further understand this, we sought to find the cellular localization of both DUBs (*Figure 5c and d*). Consistent with the recent publication, LotB specifically co-localized with the ER marker protein Calnexin, but not with other organelle markers (TOMM20 and GM130 for mitochondria and Golgi, respectively; *Figure 5c*), and the OTU domain itself failed to localize on the ER (*Figure 5—figure supplement 1a*). By contrast, we could not find a specific cellular localization of LotC (*Figure 5d*). Next, to gain more insights into the functional roles of LotB and LotC, we decided to explore combinatorial ubiquitination events with other ubiquitin-related Legionella effector proteins. To do this, we co-transfected the cells with one of the Lot-DUBs and previously known Legionella E3 ligases (SidC or SdcA) (*Hsu et al., 2014*; *Wasilko et al., 2018*). We chose these two ligases because their cellular substrates are poorly studied. TMT-labeled samples from cells expressing either SidC or SdcA alone, or together with a catalytically inactive mutant of LotB or LotC, were prepared (four combinations; SidC-LotB, SidC-LotC, SdcA-LotC, or SdcA-

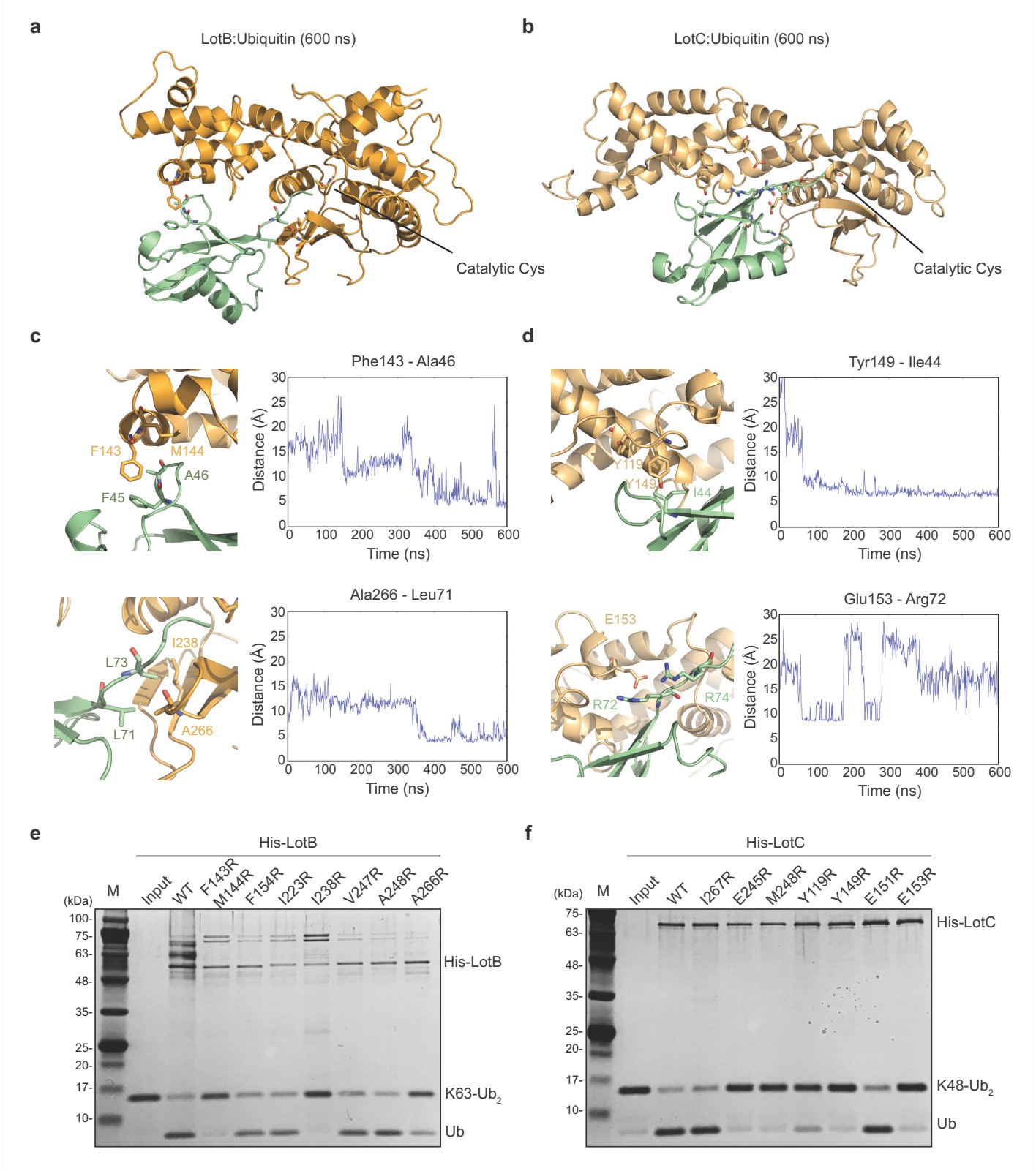

**Figure 4.** Ubiquitin-binding sites on LotB and LotC. (a, b) Molecular docking and simulations of monoubiquitin to LotB and LotC. Shown are representative snapshots of the MD simulations. Catalytic cysteine and key residues for the interaction between ubiquitin and LotB or LotC are depicted as sticks. (c, d) Key residues mediating interactions between ubiquitin and LotB or LotC. Residues are highlighted in the structure (left). Side-

*Figure 4 continued on next page*

*Figure 4 continued*

chain center-of-mass distances are shown as a function of the simulation time (right). (e, f) Di-ubiquitin cleavage assay for mutants of LotB and LotC. The catalytic activity of LotB or LotC wild-type and their mutants was tested with K63- or K48-linked Ub2, respectively.

LotC, *Figure 5—figure supplement 2a–d*). Interestingly, we identified a distinct sub-class of substrates. Overall, a smaller number of proteins were enriched with LotB compared to LotC. We reasoned that LotB specifically interacts with proteins modified with K63-linked ubiquitin chains, while LotC interacts with different types of ubiquitin chains. Intriguingly, we found a significant number of ribosome-structural proteins in LotC:SdcA combination, which were not enriched in the SidC background. Even though, our interactome study provided us useful information on putative host-interacting partners of LotB and LotC, it is still possible that only one of these proteins is genuine interactor and others are enriched through the complex formation. To avoid this question and to identify host-specific substrates of LotB or LotC under Legionella-infected condition, we developed new MS approaches. HEK293T cells were transfected with CD32 to facilitate the infection and subsequently infected with Legionella. The infected lysates were then subjected to GST pull-down with wild-type and catalytic dead mutant of LotB or LotC (*Figure 6a*). The catalytic mutant of both LotB and LotC efficiently enriched ubiquitinated proteins (*Figure 6b and d*). Interestingly, several Legionella proteins were enriched from both DUBs (*Figure 6d and e*). The LotB-C29S pulled-down some essential Legionella proteins such as atpD;ATP synthase, lpg2812;sporulation protein, lpg0841;Toluene ABC transporter, and SdhA;succinate dehydrogenase. In contrast, the LotC-C24S enriched two Legionella ribosomal proteins (rplT and rpsM) and a DNA recombinase (recA). It will be interesting to study the ubiquitination level on these Legionella proteins during infection. As expected, many of the host proteins are also enriched in these experiments. To validate whether these proteins are genuine substrate of LotB or LotC, ubiquitination level of selected substrates was analyzed by in vitro deubiquitination assay (*Figure 6g and h*). The ubiquitination level of all six substrates from different subcellular localization (RYK, Rab13, and PCYT1A for LotB, VAT1, HMOX1, and PPP2R1A for LotC) was reduced upon treatment of purified LotB or LotC, suggesting that the enriched proteins from catalytic dead mutants are putative substrates of LotB or LotC. Further studies on how the ubiquitination level of these proteins is regulated by LotB or LotC will unveil physiological roles of both LotB and LotC during Legionella infection.

## Discussion

In this study, we have identified two novel bacterial OTU-DUBs from Legionella, which we suggest to be founding members of a new sub-class of OTU-DUBs. Unlike classical OTU-DUBs, the LOT-DUBs possess extended helical insertions between the catalytic Cys-loop and the variable loop. Molecular dynamic simulations, in combination with biochemical studies, showed that the helical insertions interact with ubiquitin. As this insertion is unique for LOTs and not found in other known OTU-family members, LOT-DUBs define a new sub-class of the OTU-DUB family. We have also shown that LotB and LotC have preferences for certain ubiquitin chains and have a distinct cellular localization. Moreover, host-protein interactome studies revealed that LotB and LotC have different sets of host-interacting proteins. Together, these findings establish guidance on screening more DUBs in other pathogenic bacteria or viruses and characterizing their physiological roles during infection.

We also showed that the two LOTs have different ubiquitin-binding modes that enable them to cleave specific ubiquitin chains. With ubiquitin ABPs (Prg- and VME-probes), we showed that LotB contains an additional ubiquitin-binding site (S1') and is specific to K63-linked ubiquitin chains. In contrast, LotC cleaves various types of ubiquitin chains. Interestingly, we observed a modification of LotB with NEDD8-Prg ABP. Further studies on neddylated proteins with LotB will give us more insights into the dual-activity of LotB. In contrast, we could not see the modification between NEDD8-ABP and LotC. We reasoned that the Arg72 on ubiquitin, which is replaced by alanine in NEDD8, is essential to locate the C-terminus of ubiquitin to the catalytic site. Indeed, in molecular dynamic simulations of LotC with ubiquitin, we found Arg72 from transient interactions with Glu153 of LotC. Further structural analysis will shed light on how different ubiquitin chains bind to LotB or LotC through the different binding modes.

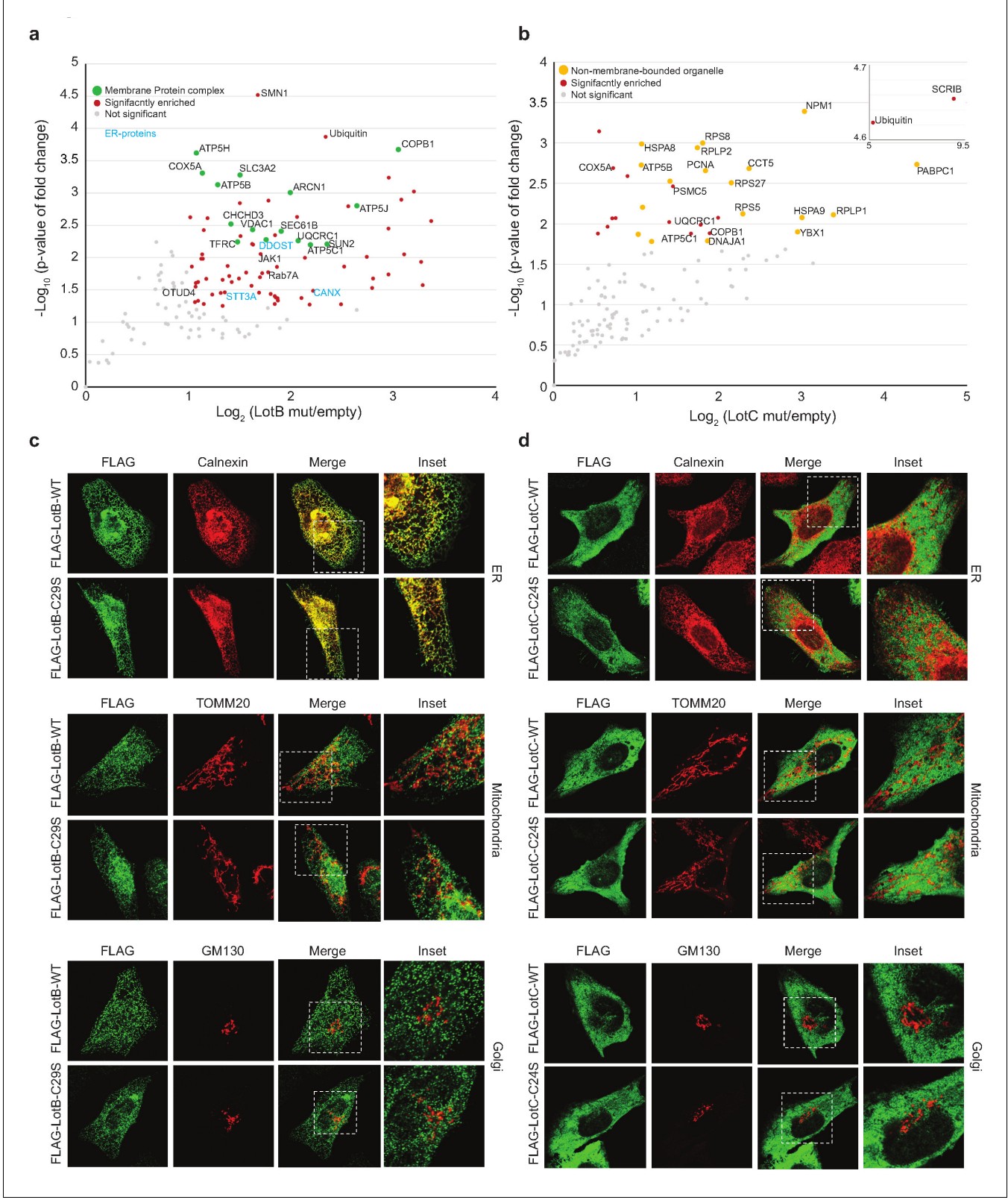

**Figure 5.** Host-interacting proteins and cellular localization of LotB and LotC. (**a, c**) Proteomic analysis of interacting partners of LotB and LotC. Catalytically inactive FLAG-LotB (C29A) and FLAG-LotC (C24A) were transfected and immunoprecipitated. Co-precipitated interacting proteins were analyzed by mass spectrometry. (**b, d**) Cellular localization of LotB and LotC. FLAG-tagged LotB and LotC were ectopically expressed in U2OS cells and immune-stained with cellular organelle markers (endoplasmic reticulum: Calnexin, mitochondria: TOMM20, Golgi: GM130).

*Figure 5 continued on next page*

*Figure 5 continued*

The online version of this article includes the following source data and figure supplement(s) for figure 5:

**Source data 1.** Mass spectrometry data used in *Figure 5a and b*.
**Figure supplement 1.** Cellular localization of LotB full-length and LotB-OTU.
**Figure supplement 2.** Proteomic analysis of interacting partners of LotB and LotC together with *Legionella* E3s.
**Figure supplement 2—source data 1.** Mass spectrometry data used in *Figure 5—figure supplement 2a, b*.
**Figure supplement 2—source data 2.** Mass spectrometry data used in *Figure 5—figure supplement 2c, d*.

DUBs from bacteria and viruses have been shown to alter the host immune system. For example, papain-like proteases (PLPro) from coronaviruses, such as middle east respiratory syndrome (MERS), severe acute respiratory syndrome (SARS) or SARS-CoV-2, have dual DUB and de-ISGylation activities and antagonized type I interferon (IFN-I) response, which is the primary defense system against viral infections (*Davis and Gack, 2015*; *Devaraj et al., 2007*; *Frieman et al., 2009*; *Sadler and Williams, 2008*). Interestingly, we identified OTUD4 as an interacting partner of LotB. OTUD4 has been shown to deubiquitinate K63-linked chains of myeloid differentiation primary response 88 (MYD88) and to downregulate NF-kB-dependent inflammation. While the recent study on LotB showed no detectable inhibition of NF-kB reporter expression (*Ma et al., 2020*), further studies are awaited to show the cross-talk between LotB and the host ubiquitination system.

*L. pneumophila* has been shown to possess multiple genes altering the host ubiquitination system. However, little is known about functional cross-talk between ubiquitin ligases and DUBs. To understand the combinatorial effects of Legionella ubiquitin ligases and DUBs, we analyzed the interactome of LotB/LotC with host proteins in the presence or absence of Legionella ligases (SidC and SdcA). We could see apparent differences in the number of enriched proteins in different combinations. A significant number of ribosomal proteins were enriched with LotC in the background of SdcA, but not from SidC background, which has 71% sequence similarity to SdcA. This finding suggests distinct physiological roles of SdcA and SidC, and a putative relationship between LotC and SdcA on regulating translation processes. Since LotC processes different types of ubiquitin chains and is mainly localized to the cytosol, the catalytically dead version of LotC can be used as a standard tool for identifying specific substrates of other known Legionella ligases. We could also nicely enrich ubiquitinated substrates of LotB and LotC from Legionella-infected cell lysates. Both DUBs enriched several Legionella proteins together with various of host proteins. It would be interesting to understand how all these ubiquitin machineries work together and alter the host-ubiquitination system at different time points of infection.

## Materials and methods

### Protein expression and purification

All proteins used in this study were expressed and purified as previously described (*Bhogaraju et al., 2016*; *Qiu et al., 2016*). Lpg1621 (LotB), Lpg2529 (LotC), Lpg2411, and Lpg2907 were cloned into either pParallelGST2 or pParallelHis2 vector (*Sheffield et al., 1999*). T7 express *Escherichia coli* competent cells (NEB) were transformed with plasmids and grown in LB medium to an $OD_{600}$ of 0.6–0.8 at 37°C. Protein expression was induced by the addition of 0.5 mM IPTG (isopropyl D-thiogalactopyranoside), and the cells were further grown overnight at 18°C and harvested. The cell pellet was resuspended in lysis buffer (50 mM Tris-HCl pH 7.5, 150 mM NaCl, and 2 mM DTT) and lysed by sonication and centrifuged at 13,000 rpm to clarify the supernatant. The supernatant of GST-tagged protein was incubated for 1 hr with glutathione-*S*-sepharose which is pre-equilibrated with washing buffer (50 mM Tris-HCl pH 7.5, 500 mM NaCl, and 2 mM DTT), and nonspecific proteins were cleared with washing. GST-proteins were eluted with elution buffer (50 mM Tris-HCl pH 8.0, 50 mM NaCl, 2 mM DTT, and 15 mM reduced glutathione) and buffer exchanged to storage buffer (50 mM Tris-HCl pH 7.5, 150 mM NaCl, and 1 mM DTT). For His-tagged proteins, the supernatant was incubated with Ni-NTA pre-equilibrated with washing buffer (50 mM Tris-HCl pH 7.5, 500 mM NaCl, and 20 mM Imidazole) for 2 hr and eluted with elution buffer (50 mM Tris-HCl pH 7.5, 500 mM NaCl, and 300 mM imidazole) and the buffer was exchanged to the storage buffer. For $LotC_{14\text{-}310}$, instead of using the elution buffer, glutathione beads were incubated with sfGFP-TEV protease

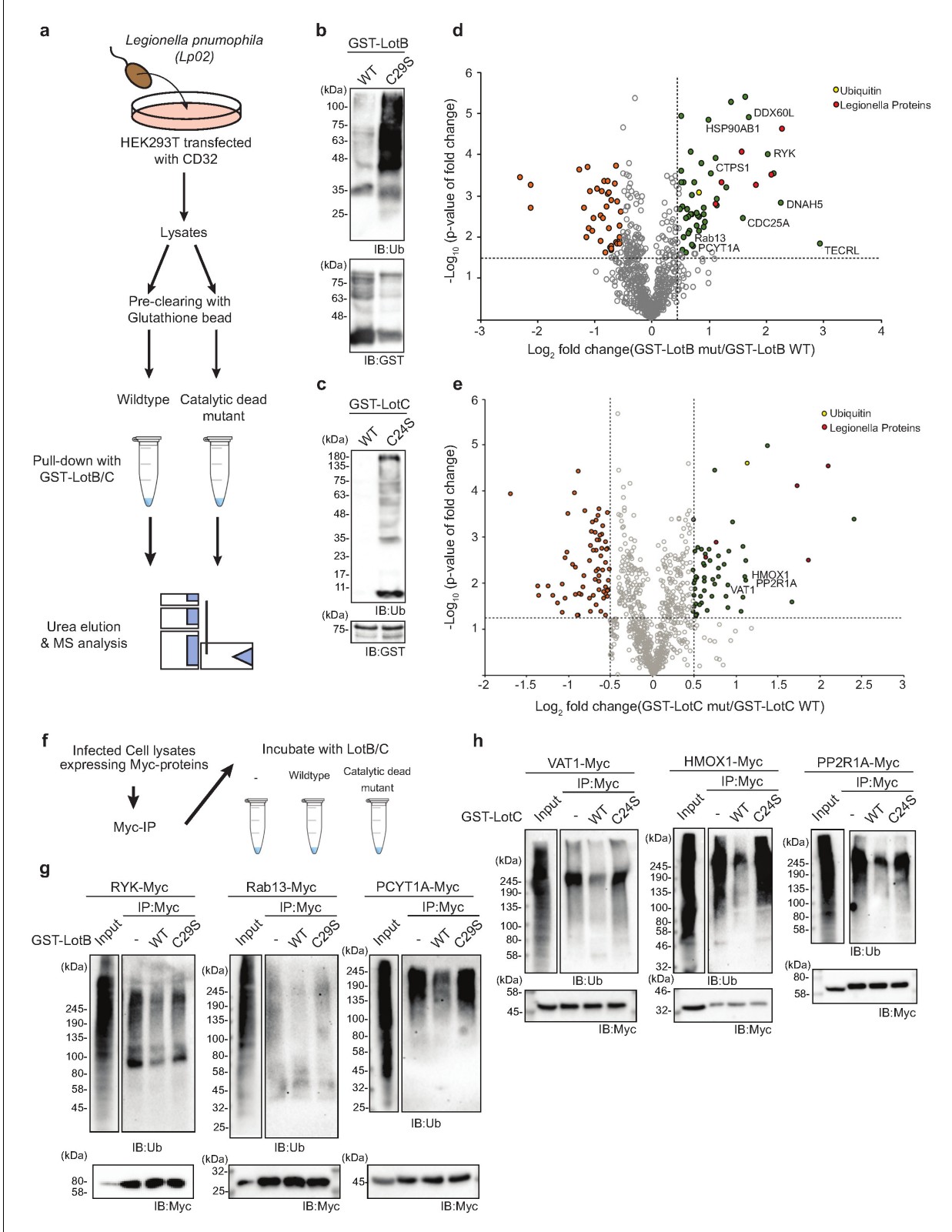

**Figure 6.** Substrate identification of LotB and LotC proteomic analysis of potential substrates of LotB and LotC. (**a–c**) Schematic of the experiment and subsequent validation using western blot. (**d, e**) Volcano plot depicting the identified proteins with corresponding fold change and p-values. Comparison was done between Mut and WT deubiquitinase (DUB). Enriched proteins with Log2 Fold change ≥ 0.5 along with −Log10 p-value ≥ 1.3 was considered for further validation. (**f–h**) Immunoprecipitation of myc from the infected lysates was performed to enrich the potential substrates for

*Figure 6 continued on next page*

*Figure 6 continued*

LotB, which are RYK, Rab13, and PCYT1A, and for LotC, which are VAT1, HMOX1, and PPP2R1A, respectively. The enriched potential substrates were further incubated with wild-type or catalytic dead mutant DUB, followed by western blotting to detect ubiquitin and myc expression.

The online version of this article includes the following source data for figure 6:

**Source data 1.** Mass spectrometry data used in *Figure 6d*.
**Source data 2.** Mass spectrometry data used in *Figure 6e*.

(*Wu et al., 2009*) overnight at 4°C. Cleaved protein was buffer exchanged to IEX buffer A (20 mM Tris-HCl pH 8.0, 20 mM NaCl, and 1 mM DTT) and purified by anion-exchange chromatography on HitrapQ (GE Healthcare) with gradient elution with IEX buffer B (20 mM Tris-HCl pH 8.0, 1 M NaCl, and 1 mM DTT) and fractions contacting samples were loaded onto size-exclusion column (Superdex 75 16/60, GE Healthcare) pre-equilibrated with 50 mM Tris-HCl pH 7.5, 50 mM NaCl, and 1 mM TCEP. Proteins were concentrated to 20 mg/ml and stored for crystallization.

## Di-Ub panel cleavage assay

To activate DUBs, 3 µl of 5 µM of DUBs were mixed with 12 µl of activation buffer (25 mM Tris-HCl pH 7.5, 150 mM NaCl, and 10 mM DTT) and incubated 15 min at 25°C. For di-ubiquitin samples, 3 µl of di-ubiquitin chains (1 mg/ml) were mixed with 3 µl 10× reaction buffer (500 mM Tris-HCl pH 7.5, 500 mM NaCl, and 50 mM DTT) and 12 µl of ultra-pure water. To initiate the reaction, the activated DUBs were mixed with di-ubiquitin, and samples were taken at the indicated time points. The reactions were quenched by the addition of SDS-sample buffer. The samples were further analyzed by SDS-PAGE and stained with a silver-staining kit (Pierce Silver Staining Kit, Thermo Fischer).

## Ubiquitin/NEDD8/SUMO-/ISG15/UFM ABPs assay

DUBs were diluted (1.5 µM, final concentration) with activation buffer and incubated 10 min at 25°C and the ABPs were diluted (50 µM, final concentration) in dilution buffer (50 mM Tris-HCl pH 7.5 and 150 mM NaCl). A total of 30 µl of the reaction mixture was prepared by mixing 20 µl of activated DUBs (1.5 µM), 3 µl of ABPs, 3 µl of reaction buffer (500 mM Tris-HCl pH 7.5, 500 mM NaCl, and 50 mM DTT), and 4 µl of ultra-pure water. Samples were taken at the indicated time points, and the reactions were quenched by the addition of SDS-sample buffer. Samples were further analyzed by SDS-PAGE and stained with Coomassie staining solution.

## Crystallization

The concentrated LotC$_{14-310}$ were screened with sitting drop matrix screens in a 96-well plate with 100 nl of protein and 100 nl of precipitant solution at 293 K. Initial crystals appeared from solution containing 25% PEG3350, 100 mM Tris-HCl pH 8.5, and 200 mM NaCl with 18.4 mg/ml protein concentration. Diffraction-quality crystals were grown in optimized solution containing 19% PEG 3350, 100 mM Tris-HCl pH 8.5, and 150 mM NaCl with 24 mg/ml protein concentration.

## Data collection, processing, and structure determination

To obtain the phase, 0.4 µl of 10 mM K$_2$PtCl$_4$ was added to the drop containing crystals and incubated for 18 hr. Heavy atom-soaked crystals were cryo-protected using mother liquor solution supplemented with 15% (v/v) glycerol. Diffraction data were collected on a single frozen crystal in a nitrogen stream at 100 K at beamline PXI as Swiss Light Source, Villigen. Initial data sets were processed using XDS (*Kabsch, 2010*), and initial-phases were determined by Autosol in Phenix (*Terwilliger et al., 2009*). Structure refinement and manual model building were performed with Coot and Phenix. Refine (*Afonine et al., 2012*; *Emsley et al., 2010*).

## Protein–protein docking

We used the Rosetta protein–protein docking method (*Gray et al., 2003*) to identify low-energy conformations of the complexes of ubiquitin with LotB and LotC. Given that the C-terminus of ubiquitin should interact with the catalytic residue of the OTUs, we used the local docking approach in which we placed the C-terminal end of ubiquitin (Gly76) near the catalytic residues in both ligases

(Cys29 and His270 for LotB and Cys16 and His296 for LotC). We then started the docking by optimizing the rigid-body orientation and side-chain conformation sampling. The program requires two protein structures as inputs, which were prepared by running the refinement protocol before the docking step. We performed the local docking approach and generated 100 independent structures for each complex. The complexes in this way were subject to local refinement to remove remaining small clashes. The complexes were then clustered based on the distance matrix of Cα atoms between the ligase and ubiquitin using the KMeans method. The representatives of two major clusters in each case were selected based on the interface score (I_sc), which represents the energy of the interactions across the interface of two proteins. These representative complexes were used for MD simulations.

## Molecular dynamics simulations

All-atom explicit solvent molecular dynamics (MD) simulations were performed for two docking results for each ligase. The systems were built using the CHARMM-GUI web server (*Wu et al., 2014*). The systems were hydrated with 150 mM NaCl electrolyte. The all-atom CHARMM36m force field was used for proteins, lipids, and ions, and TIP3P was used for water molecules (*Best et al., 2012*). The MD trajectories were analyzed with visual molecular dynamics (VMD) (*Humphrey et al., 1996*). The MD simulations were performed using GROMACS 2019 (*Abraham et al., 2015*). The starting system was minimized for 5000 steps with the steepest descent energy minimization and equilibrated for 6.5 ns of MD simulation first in the NVT ensemble (1.5 ns) and then in the NPT (5 ns) ensemble, in which all non-hydrogen atoms of the protein were restrained to the fixed reference positions with progressively reduced force constants, starting at 1000 kJ/mol·nm$^2$. Afterwards, the production runs were carried out in the NPT ensemble for 600 ns for each setup. To keep the C-terminus of ubiquitin in the catalytic site, 7 Å wall restraints were placed on the distance between Cα of G76$_{UB}$ and Cys29/His270 in LotB and between Cα of G76$_{UB}$ and Cys16/His296 in LotC. Periodic boundary conditions were used. Particle mesh Ewald (*Darden et al., 1993*) with cubic interpolation and 0.12 nm grid spacing for Fast Fourier Transform was used to treat long-range electrostatic interactions. The time step was two fs. The LINCS algorithm (*Hess et al., 1997*) was used to fix all bond lengths. Constant temperature (310 K) was set with a Nosé-Hoover thermostat (*Hoover, 1985*), with a coupling constant of 1.0 ps. An isotropic Parrinello-Rahman barostat (*Parrinello and Rahman, 1981*) with a coupling constant of 5.0 ps was used to maintain a pressure of 1 bar.

## Identification of host-interacting proteins of LotB and LotC

For interactome analysis, HEK 293 cells were transfected with FLAG-LotB WT or FLAG-LotB C29A and FLAG-LotC WT or FLAG-LotC C24A. To identify the substrates or interactors modified by Legionella-derived E3 ligases, GFP-SidC or GFP-SdcA were co-transfected with FLAG-LotB WT or FLAG-LotB C29A and FLAG-LotC WT or FLAG-LotC C24A. Three independent biological replicates were processed per experiment for downstream statistical analysis. Since in some instances comparing between Mut over Wt DuB did not enrich ubiquitin significantly we looked for interacting partners by comparing between Mut over empty vecvotr. Cells were lysed in ice cold lysis buffer (50 mM Tris-HCl, pH 7.5; 150 mM NaCl; 1% Triton x-100) and equal amount of lysates were incubated with FLAG-M2 beads in IP buffer (Lysis buffer without detergent). After incubation, IPs were washed with wash buffer (50 mM Tris-HCl, pH 7.5; 400 mM NaCl; and 0.5 mM EDTA) and the interacting proteins were eluted with 8 M urea solution. After the reduction and alkylation with TCEP and chloroacetamide, the samples were digested with 0.5 µg trypsin (Promega) at 37°C overnight after diluting the urea <2 M. Digests were acidified using trifluoroacetic acid (TFA) to a pH of 2–3, and the peptides were enriched using C18 stage tips (*Rappsilber et al., 2003*). To get quantitative information, peptides were either labeled with TMT 10 plex reagent (Thermo fisher) or analyzed label-free. The peptides were separated on an in-house made C18 column (20 cm length, 75 µm inner diameter, and 1.9 µm particle size) by an easy n-LC 1200 (ThermoFisher) and directly injected in a QExactive-HF or in case of TMT samples into a Fusion Lumos mass-spectrometer (ThermoFisher) and analyzed in data-dependent mode. Data analysis was done using Maxquant 1.65 (*Cox and Mann, 2008*). Fragment spectra were searched against *Homo sapiens* SwissProt database (TaxID:9606, version 2018). Label-free quantification was done with MaxLFQ (*Cox et al., 2014*) method with activated match between runs. TMT-labeled samples were analyzed by using TMT 10 Plex option within the software.

Further normalization was done using NormalyserDE (*Willforss et al., 2019*). Statistically significant changes between samples were determined using a two-sample t-test with a permutation-based FDR of 5% on log2 transformed values in Perseus (*Tyanova et al., 2016*). Data files are available in supplementary files.

## Identification and validation of putative substrates of LotB and LotC

For identifying potential substrates, we performed GST pull down for both the proteins from infected lysates. For proteomic-based identification, three independent biological replicates were processed. To this end, HEK293T cells were transfected with CD32 and infected with Legionella WT strain for 2 hr. Cells were lysed with ice cold lysis buffer (10 mM Tris-HCl pH 7.5, 150 mM NaCl, 0.5 mM EDTA pH 8, and 0.5% NP40). Total 1 mg lysate was used for IP. Lysates were incubated with glutathione beads to rule out background binding and the precleared lysates were incubated with fresh bead containing pure GST inactive DuB protein in 1:100 ratio (pure DuB:Lysate). IPs were washed three times with wash buffer (10 mM Tris-HCl pH 7.5, 300 mM NaCl, and 0.5 mM EDTA pH 8) and subsequently two times with IP buffer (Lysis buffer without detergent) and two times with MS grade water prior to urea elution. Samples were processed for mass spectrometry as described in previous section. Data files are available in supplementary files.

For validating the potential substrates from the results of mass spectrometry, we performed in vitro deubiquitination assay for selected targets. HEK293T cells were first transfected with myc-tag proteins and CD32, followed by Legionella WT strain infection for 2 hr. Cells were lysed with ice cold lysis buffer (50 mM Tris-HCl pH 7.5, 150 mM NaCl, 5 mM EDTA pH 8, and 1% NP40). Total 2 mg lysate was incubated with myc agarose (Sigma Aldrich) at 4°C, overnight. After washing three times with wash buffer (50 mM Tris-HCl pH 7.5, 400 mM NaCl, and 5 mM EDTA pH 8), the agarose beads were incubated with pure GST wild-type or catalytic dead mutant DUB protein at 37°C for 1.5 hr and subsequently washed by wash buffer for two times. Samples were boiled in sample buffer and further detected the ubiquitination level by western blotting.

## Confocal imaging and image analysis

U2OS cells were transfected with FLAG-LotB or FLAG-LotC by GeneJuice transfection reagent (Merck) for 24 hr and fixed by 4% paraformaldehyde for 20 min. After fixation, cells were permeabilized and blocked by 0.1% saponin and 1% BSA in PBS for 1 hr at room temperature. Cells were incubated with anti-Flag antibody (Sigma and Cell Signaling), with either anti-calnexin antibody (Abcam), anti-TOMM20 antibody (Abcam), or anti-GM130 (BD Transduction Laboratories) at 4°C overnight. Alexa Fluor 488 and Alexa Fluor 546 (Invitrogen) secondary antibody were incubated for 1 hr at room temperature. Images were acquired by the Zeiss LSM780 microscope system with $63 \times 1.4$ NA oil immersion objective and further analyzed by Zeiss Zen microscope software.

## Cell lines

HEK293T (ATCC CRL-3216) and U2OS (ATCC HTB-96) were used in this study. Both cell lines were authenticated by STR profiling from the suppliers (ATCC). Cells were tested negative for mycoplasma contamination.

## Acknowledgements

We thank Yuxin-Mao for providing SidC and SdcA clones. We also thank Andrea Gubas for critical reading and Stefan Knapp for the advice in structure determination and sharing synchrotron time. The authors also thank the staff at SLS for their support during crystallographic X-ray diffraction data collection. The data collection at SLS has been supported by the funding from the European Union's Horizon 2020 research and innovation program under grant agreement number 730872, project CALIPSOplus. This project was supported by the European Research Council (ERC) under the European Union's Horizon 2020 research and innovation program (ID, grant agreement No 742720), the LOEWE program DynaMem of the State of Hesse (Germany, Project-ID III L6-519/03/03.001 – [0006]), and Deutsche Forschungsgemeinschaft (DFG, German Research Foundation Project-ID 259130777 – SFB1177; Leibniz-Program to ID; CEF-MC - EXC115/2; SFB 902), the Max Planck Society and NWO-VIDI grant and Off-rad grant for G.H.

## Additional information

### Competing interests

Ivan Dikic: Reviewing editor, *eLife*. The other authors declare that no competing interests exist.

### Funding

| Funder | Grant reference number | Author |
|---|---|---|
| European Research Council | 742720 | Donghyuk Shin<br>Anshu Bhattacharya<br>Yi-Lin Cheng<br>Marta Campos Alonso<br>Ivan Dikic |
| Deutsche Forschungsge-meinschaft | ID 259139777-SFB1177 | Ivan Dikic |
| LOEWE program DynaMem of the State of Hesse | Project-ID III L6-519/03/03.001 –(0006) | Ivan Dikic |
| German Research Foundation | (Project-ID259130777 – SFB1177 | Ivan Dikic |
| Leibniz Association | | Ivan Dikic |
| Max Planck Society | | Ahmad Reza Mehdipour<br>Gerhard Hummer |
| LOEWE program of the State of Hesse | DynaMem | Gerhard Hummer |
| Deutsche Forschungsge-meinschaft | SFB1177 | Ahmad Reza Mehdipour<br>Gerhard Hummer |
| ZonMw | 451001026 | Gerbrand J van der Heden van Noort |
| NWO | VI.Vidi.192.011 | Gerbrand J van der Heden van Noort |
| Deutsche Forschungsge-meinschaft | CEF-MC-EXC115/2 | Ahmad Reza Mehdipour<br>Gerhard Hummer |
| Deutsche Forschungsge-meinschaft | SFB902 | Ahmad Reza Mehdipour<br>Gerhard Hummer |

The funders had no role in study design, data collection and interpretation, or the decision to submit the work for publication.

### Author contributions

Donghyuk Shin, Conceptualization, Validation, Investigation, Writing - original draft, Writing - review and editing, Solving structure, biochemical and biophysical assays; Anshu Bhattacharya, Data curation, Formal analysis, Designed, performed and analyzed the Mass spectrometry experiments; Yi-Lin Cheng, Marta Campos Alonso, Gerbrand J van der Heden van Noort, Formal analysis; Ahmad Reza Mehdipour, Formal analysis, Investigation; Huib Ovaa, Supervision; Gerhard Hummer, Supervision, Writing - review and editing; Ivan Dikic, Conceptualization, Supervision, Funding acquisition, Project administration, Writing - review and editing

### Author ORCIDs

Donghyuk Shin https://orcid.org/0000-0002-8272-6133
Anshu Bhattacharya https://orcid.org/0000-0002-7383-3594
Yi-Lin Cheng https://orcid.org/0000-0002-3603-5913
Gerhard Hummer https://orcid.org/0000-0001-7768-746X
Ivan Dikic https://orcid.org/0000-0001-8156-9511

Decision letter and Author response
Decision letter https://doi.org/10.7554/eLife.58277.sa1
Author response https://doi.org/10.7554/eLife.58277.sa2

## Additional files

### Supplementary files
- Supplementary file 1. Data collection and refinement statistics table.

- Transparent reporting form

### Data availability

Diffraction data have been deposited in PDB under the accession code 6YK8.

The following dataset was generated:

| Author(s) | Year | Dataset title | Dataset URL | Database and Identifier |
|---|---|---|---|---|
| Shin D, Dikic I | 2020 | OTU-like deubiquitinase from Legionella- Lpg2529 | https://www.rcsb.org/structure/6YK8 | RCSB Protein Data Bank, 6YK8 |

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
