## [Decision Letter]

**Acceptance summary:**

This paper describes a biochemical analysis of LotC and LotB, two OTU domain DUBs found in *Legionella*. The work includes a determination of the substrate specificity of the DUBs on ubiquitin conjugates, as well as a structural analysis of LotB and LotC. The paper also examines interaction partners for Lot proteins, as well as their subcellular localization. The interaction partners for the DUBs provides a resource for the community. Your efforts to address the previous reviewers comments made the paper much stronger.

**Decision letter after peer review:**

Thank you for submitting your article "Novel class of OTU deubiquitinases regulate substrate ubiquitination upon *Legionella* infection" for consideration by *eLife*. Your article has been reviewed by three peer reviewers, one of whom is a member of our Board of Reviewing Editors, and the evaluation has been overseen by Cynthia Wolberger as the Senior Editor. The following individual involved in review of your submission has agreed to reveal their identity: Claudio A P Joazeiro (Reviewer #1).

The reviewers have discussed the reviews with one another and the Reviewing Editor has drafted this decision to help you prepare a revised submission.

Summary:

This paper describes a biochemical analysis of LotC and LotB, two OTU domain DUBs found in *Legionella*. The work includes a determination of the substrate specificity of the DUBs on ubiquitin conjugates, as well as a structural analysis of LotB and LotC. The paper also examines interaction partners for Lot proteins, as well as their subcellular localization. The interaction partners for the DUBs provides a resource for the community.

Essential revisions:

While the reviewers feel that the work is potentially appropriate for publication in *eLife*, they feel that additional work is needed to strengthen various aspects of the paper.

1) Given that LotB is already known to be an OTU DUB, careful and appropriate descriptions of what is known and what is new would strengthen the paper. In addition, it would be helpful to spend more time presenting the unique features of LotC relative to LotA/B. Are there new mutations that would be easy to make and test that would lead to some validation of the unique features of LotC? Perhaps you already have such data. In this regard, it would be interesting to compare a simple docking of Ub to the molecular dynamics simulation.

2) The reviewers feel that the interaction proteomics is less well developed than it might be. Recognizing that it may be difficult to develop novel biological insights in the current situation, the reviewers feel that more effort in organizing the data would improve the paper substantially. Many of the interactors are in complexes and so potentially only one protein within the complex is a substrate. So the description and interpretation of the data should also be more precise. The reviewers feel that some additional form of orthogonal validation would really help. Is it possible to try to validate at least s small number of the potential interactions? A second possibility is to take perhaps a small number of potential high value targets (like Rabs, given that they are known to be ubiquitylated in the context of *Legionella* infection), and see if their ubiquitylation is reduced by LotB/C expression using a simple western blot assay. Other likely ubiquitylated proteins could also potentially be tested in a similar way. In addition, the interaction with the ribosome is interesting but given its abundance, raises the question of whether it could be a false positive. Can this interaction be validated better? Organizing the hits into protein complexes and presenting the data as a "complexome" might help.

3) The interaction data is provided only for the inactive Lot proteins. If available, this data should also be provided for the WT protein. Also, the localization data is provided for the WT protein, so in order to be parallel with the proteomics, it would also be important to have the mutant protein localization data as well.

Because some of the comments of the reviewers may facilitate an understanding of the desired revisions, I have provided all of the comments below.

*Reviewer #1:Legionella pneumophila*, a Gram-negative bacterium, is an important human pathogen. Its pathogenic mechanism relies on the secretion of effector proteins into the host cell to modify innate immunity and metabolism. There is therefore great interest in identifying and characterizing such bacterial effectors. Here, Dikic and colleagues present a multi-disciplinary study of two novel *Legionella* effectors that act as deubiquitinating enzymes (DUBs) that interfere with the host ubiquitin signaling.Using a well-conceived bioinformatics approach to fish for candidate DUBs among 305 predicted secreted *Legionella* proteins, the authors found two novel DUBs belonging to the OTU family, which they named LotB and LotC. OTU-family DUBs are unique in that they can act in a ubiquitin linkage-specific manner. Indeed, results of a di-ubiquitin cleavage assay suggested that LotB acted more effectively towards Ub K63 linkages (and to a weaker extent towards K48 and K6 linkages), whereas LotC did not show linkage specificity. To elucidate the ubiquitin specificity and catalytic mechanism of these DUBs the authors used a combination of activity-based probes, structure-function studies, and crystallographic structural determination. A main feature that has emerged is that the Ub K63 linkage specificity of LotB relies on an additional ubiquitin-binding site (S1'). The authors also performed initial studies aimed at understanding the functional role of LotB and LotC in the host cell. LotB appears to localize predominantly to the ER membrane and its interactome indeed revealed primarily membrane proteins. On the other hand, LotC appears to have a broad subcellular distribution and did not exhibit preference towards membrane interactors. Interestingly, co-expression of the DUBs with selected *Legionella*-secreted E3 ligases uncovered further interactors. Among those, the combination of LotC and SdcA revealed the interaction with a number of ribosomal proteins. In conclusion, this manuscript integrates data obtained using a wide variety of experimental approaches to characterize the specificity, mechanism, structure and potential biological functions of two novel effector proteins of an important bacterial pathogen. The work will be of broad interest and merits publication.

The only major weakness of the work is the writing. First, there are many errors throughout the text ("*Legionella pneumophila* that causes Legionaries' disease"; "host-protein interactome studies revealed that LotB and LotC have different sets of host-interacting proteins to LotB and LotC"; etc). Second, communicating the manuscript's beautiful messages to a general audience would benefit enormously from a major revision to improve on explanations (e.g., S1' is not defined anywhere) and clarity overall.

Although the findings open several future directions outside the scope of the manuscript (e.g., what is the LotB membrane-targeting mechanism?) it would be exciting to see some of the functional studies developed further. For example, what are functional consequences of the ribosome interactions found? Does expression of a ubiquitin K63R mutant interfere with LotB (but not LotC) interactions?

The legends are fairly undescriptive and incomplete. For example, in Figure 5E-F, what is the basis for classification of the different protein groups? I was particularly confused by the presence of proteins such as CNOT1, HNRNPs, PCNA, PSMCs, PSMDs, SMN1, etc., listed under "Orgenelle" in Figure 5E. And what is the light green box in Figure 5E?

Reviewer #2:

This paper describes a molecular analysis of LotB and LotC DUBs from Legionelle, and defines these enzymes as well as the previously characterized LotA DUB as unique OTU-like DUBs. The paper combines a lot of biochemical analysis of substrate specificity with structural biology to elucidate mechanisms underlying the interesting substrate specificities of the enzymes. LotB appears quite specific for K63 linkages whereas LotC is a bit more promiscuous and cleaves multiple chain types. Lot enzymes were found to have unique structural features in their S1 binding site compared with other OTU-class DUBs which may explain their specificity, in part. The authors used molecular dynamics to identify potential ubiquitin S1 binding site residues and tested candidate residues in vitro using chain cleavage assays. These studies verified a subset of residues as being important.

Finally, the authors perform a series of exploratory studies to understand potential biological functions of LotB and LotC. The authors demonstrate that LotB is associated with ER and appears to interact with proteins that are often associated with ER, including some membrane proteins. In contrast, LotC doesn't seem to have a specific subcellular localization and interacts preferentially with soluble proteins or protein complexes. They performed analogous experiments with over expression of *Legionella* E3s, identifying additional candidate substrates.

Overall, the paper covers a lot of ground. The structural and biochemical studies are very complete. The target identification studies are a good start, but the study doesn't have much in the way of validation currently. However, this work provides many new leads for potential biological functions of the DUBs.

Reviewer #3:

This manuscript concerns the characterization of two *Legionella* OUT-like deubiquitinases LotB and LotC. The authors use a combination of activity based suicide probes and endpoint assays using di-ubiquitin to define the UBL selectivity and ubiquitin chain specificity of these enzymes. A crystal structure of the catalytic domain of LotC is presented and compared with the already existing PDB structure of LotB. The authors identify extended insertions in the catalytic domain that based on molecular dynamics simulations contribute to the mode of ubiquitin binding. The authors then conduct a series of differential proteomic interactome studies using catalytically inactive mutants as baits. Some of these experiments are conducted in the presence of overexpressed *Legionella* ubiquitin E3 ligases. Interacting proteins are not further validated but presented as likely substrates of these enzymes.

Whilst this is in principle an interesting topic, the study at this stage does not provide substantial fundamental new insights or present clear conceptual advances in the mechanistic understanding of DUBs in general or in the role of these particular DUBs in the context of a *Legionella* infection.

1) LotB (as the authors also state) was recently (in January 2020) described in a manuscript by Ma et al. (Ma et al., 2020) that included a crystal structure identifying LotB as OTU-related and also described its of K63-ubiquitin chain selectivity. Related LotA was already characterised as a DUB in 2018 (Kubori et al., 2018). Given that this study does not undertake a full blown phylogenetic/bioinformatics analysis of the family, the title seems somewhat misleading. This manuscript is not the first to describe the "novel class of OTU deubiquitinases". The biochemical characterization of the enzymatic activity is quite basic and the cell biology of the enzymes is not explored except for an attempt at subcellular localization which identifies LotB as an ER-localised protein.

2) Interactors are not necessarily substrates – Whilst the title alludes to substrate ubiquitination, and the Introduction states "we…identified the specific host-substrates of LotB and LotC", the experiments are not directed to identify the substrates – i.e. proteins that may be less ubiquitinated in the presence of the *Legionella* DUBs. This would have required a different approach, for example isolating ubiquitylated proteins or peptides in the first place.

The interactome datasets may of course identify interesting binding partners that may well include both regulators and substrates. However no orthologous validation experiments have been conducted to verify firstly the interaction and secondly assess the putative enzyme-substrate relationship. The figures focus on the comparison of interactors that are enriched in "mutant over empty" – presumably an empty plasmid – rather than drawing a comparison between wild-type and inactive mutants. Importantly, there is a priori no evidence that catalytically inactive DUBs would act as true substrate traps – they may misfold or mislocalise and for that reason associate non-specifically with a different subset of proteins. Thus in the absence of further work, the information value of these datasets is unclear.

The introduction of some *Legionella* E3 ligases in the experimental set-up does alter the interactome data but this could be due to an overall change in the ubiquitin landscape upon the overexpression of those E3s. In other words, overexpression of any other E3 may also impinge on the interactome associated with the *Legionella* DUBs. This does not necessarily imply a coordination of activities in the host. The overall rationale of those experiments is not clear – the *Legionella* E3 ligases do not necessarily modify host proteins that are then targeted by *Legionella* DUBs.

---

## [Author Response]

Essential revisions:While the reviewers feel that the work is potentially appropriate for publication in eLife, they feel that additional work is needed to strengthen various aspects of the paper.1) Given that LotB is already known to be an OTU DUB, careful and appropriate descriptions of what is known and what is new would strengthen the paper. In addition, it would be helpful to spend more time presenting the unique features of LotC relative to LotA/B. Are there new mutations that would be easy to make and test that would lead to some validation of the unique features of LotC? Perhaps you already have such data. In this regard, it would be interesting to compare a simple docking of Ub to the molecular dynamics simulation.

We highly appreciate these specific comments. We agree that the structure and the chain specificity of Lpg1621 (Ceg23) was published in January 2020 while we were studying both LotB (Lpg1621;Ceg23) and LotC. We also agree that Kubori et al., introduced the name *Legionella* OTU deubiquitinase (LOT) for the first time in 2018 (Kubori et al., 2018). However, while both manuscripts described Lpg1621(Ceg23) and LotA as OTU-related deubiquitinases, there was no clear description of *Legionella* OTU-deubiquitinase as unique DUB family. In this manuscript, we named Lpg1621(Ceg23) as LotB and Lpg2529 as LotC to be consistent with LotA and put all of them as unique OTU subfamily based on their structures. We spotted that all three *Legionella* OTU-deubiquitinases (LotA, LotB and LotC) are having an extra insertion between Cys-loop and variable loop and suggested this insertion as unique feature of *Legionella* OTU-deubiquitinase family.

The ubiquitin docking MD simulation is presented in Figure 4, based on this we found key mutations on LotC (E245R, M248R, Y119R, Y149R, E153R, Figure 4B, D, F) that result in the failure of K48-Ub2 cleavage. Importantly, this mutation analysis also proved that the interaction between ubiquitin and LotC is mainly mediated by the ionic interaction between R72/74 of ubiquitin and E153 of LotC. This result also explains the reason why LotC would not interact with Nedd8 which lack the R72 at the C-terminal tail (Figure 2—figure supplement 1C). We also identified two additional single mutants (I238R, A266R) which impaired the K63-Ub2 cleavage activity of LotB. These mutants lead us to identify additional hydrophobic interaction residues that were not described previously (Ma et al., 2020).

2) The reviewers feel that the interaction proteomics is less well developed than it might be. Recognizing that it may be difficult to develop novel biological insights in the current situation, the reviewers feel that more effort in organizing the data would improve the paper substantially. Many of the interactors are in complexes and so potentially only one protein within the complex is a substrate. So the description and interpretation of the data should also be more precise. The reviewers feel that some additional form of orthogonal validation would really help. Is it possible to try to validate at least s small number of the potential interactions? A second possibility is to take perhaps a small number of potential high value targets (like Rabs, given that they are known to be ubiquitylated in the context of Legionella infection), and see if their ubiquitylation is reduced by LotB/C expression using a simple western blot assay. Other likely ubiquitylated proteins could also potentially be tested in a similar way. In addition, the interaction with the ribosome is interesting but given its abundance, raises the question of whether it could be a false positive. Can this interaction be validated better? Organizing the hits into protein complexes and presenting the data as a "complexome" might help.

We thank reviewers for specific comments. As reviewers concerned, interacting proteins identified from LotB or LotC be false positives due to the overexpression of both enzymes and there were indeed many proteins from complexes. To avoid such confusion, we have now clearly mentioned the possibility of false-positive hits in the manuscript and we also removed Figure 5E, F which was also the specific concerns from reviewer #1. In addition, we are now providing new MS data for substrates identification and validation in entire new figure (Figure 6). We developed new MS approaches to identify host-specific substrates of LotB or LotC under *Legionella* infected condition. HEK293T cells were infected with *Legionella* and the infected lysates were then subjected to GST pull-down with wild-type and catalytic dead mutant of LotB or LotC (Figure 6A). Interestingly, the catalytic mutant of both LotB and LotC efficiently enriched ubiquitinated proteins (Figure 6B, C). As reviewers recommended, we chose some of the hits, which have potential values from this new MS list and they are further validated by in vitro deubiquitination assay (Figure 6G, H). For Lot B, we choose three proteins, which contain critical cellular function, including receptor like tyrosine kinase (RYK), Rab13, and phosphocholine cytidylyltransferase A (PCYT1A). For Lot C, we also choose three proteins based on the MS scores and the potential function, including vesicle amine transport 1 (VAT1), heme oxygenase 1 (HMOX1), and protein phosphatase 2 scaffold subunit alpha (PPP2R1A). These substrates of LotB and Lot C shed the new light on *Legionella* pathogenesis, but still need to be further investigated.

3) The interaction data is provided only for the inactive Lot proteins. If available, this data should also be provided for the WT protein. Also, the localization data is provided for the WT protein, so in order to be parallel with the proteomics, it would also be important to have the mutant protein localization data as well.

We thank editor for asking this specific control experiment. We have now included mutant protein localization in Figure 5C, D.

Because some of the comments of the reviewers may facilitate an understanding of the desired revisions, I have provided all of the comments below.Reviewer #1:[…] The only major weakness of the work is the writing. First, there are many errors throughout the text ("Legionella pneumophila that causes Legionaries' disease"; "host-protein interactome studies revealed that LotB and LotC have different sets of host-interacting proteins to LotB and LotC"; etc). Second, communicating the manuscript's beautiful messages to a general audience would benefit enormously from a major revision to improve on explanations (e.g., S1' is not defined anywhere) and clarity overall.

We thank the reviewer for this specific comment. We have upgraded the entire text and added few additional parts and corrected identified errors across the manuscript.

Although the findings open several future directions outside the scope of the manuscript (e.g., what is the LotB membrane-targeting mechanism?) it would be exciting to see some of the functional studies developed further. For example, what are functional consequences of the ribosome interactions found? Does expression of a ubiquitin K63R mutant interfere with LotB (but not LotC) interactions?

We thank reviewer for this specific comment. We agree that exploring some of the functional consequences of interacting partner of LotB and LotC are exciting questions. Since all reviewers were concerned about possible misleading of our interactome data, we decided to develop new MS approach to find substrates of LotB or LotC (Figure 6). We prepared *Legionella* infected lysates to induce and follow the changes in ubiquitination system under infection condition and pull-downed the ubiquitinated proteins by catalytic dead version of both LotB and LotC. With this new approach, we could nicely enrich ubiquitinated substrates (Figure 6A-E) and validated some of them by in vitro ubiquitination assay (Figure 6G, H). As we also described in the Discussion, there are many interesting open questions to be followed based on our MS results. For instance, we found several *Legionella* proteins as putative substrates of LotB and LotC. It will be important to initiate follow-up studies on how host or *Legionella* E3 ubiquitinates those substrates and how LotB and LotC revert these modifications.

The legends are fairly undescriptive and incomplete. For example, in Figure 5E-F, what is the basis for classification of the different protein groups? I was particularly confused by the presence of proteins such as CNOT1, HNRNPs, PCNA, PSMCs, PSMDs, SMN1, etc., listed under "Orgenelle" in Figure 5E. And what is the light green box in Figure 5E?

We removed both Figure 5E and F in the revised manuscript and all the figure legends were upgraded.

Reviewer #3:[…] Whilst this is in principle an interesting topic, the study at this stage does not provide substantial fundamental new insights or present clear conceptual advances in the mechanistic understanding of DUBs in general or in the role of these particular DUBs in the context of a Legionella infection.1) LotB (as the authors also state) was recently (in January 2020) described in a manuscript by Ma et al. (Ma et al., 2020) that included a crystal structure identifying LotB as OTU-related and also described its of K63-ubiquitin chain selectivity. Related LotA was already characterised as a DUB in 2018 (Kubori et al., 2018). Given that this study does not undertake a full blown phylogenetic/bioinformatics analysis of the family, the title seems somewhat misleading. This manuscript is not the first to describe the "novel class of OTU deubiquitinases". The biochemical characterization of the enzymatic activity is quite basic and the cell biology of the enzymes is not explored except for an attempt at subcellular localization which identifies LotB as an ER-localised protein.

We appreciate reviewer’s comments. As reviewer pointed out, structure and the chain specificity of Lpg1621 (Ceg23) was published in January 2020 while we were studying both LotB and LotC. We also agree that Kubori et al., introduced the name *Legionella* OTU deubiquitinase for the first time in 2018 (Kubori et al., 2018). While both manuscripts described LotA and Lpg1621(Ceg23) as OTU-related deubiquitinases, there was no clear description of *Legionella* OTU-deubiquitinase as unique DUB family with specific characteristic. In this manuscript, we named Lpg1621(Ceg23) as LotB and Lpg2529 as LotC to be consistent with LotA and put all of them as unique OTU subfamily based on their structures. We spotted that all three *Legionella* OTU-deubiquitinases (LotA, LotB and LotC) are having an extra insertion between Cys-loop and variable loop and suggested this insertion as unique feature of *Legionella* OTU-deubiquitinase family. Nevertheless, we agree with the reviewer that the title seems somewhat misleading and we have corrected the title. In addition, we are now providing new proteomic data where we show substrates for both LotB and LotC with orthogonal validation. This will open many questions related to bacterial DUBs and their functions in host-cells.

We also identified two additional single mutants (I238R, A266R) which impaired the K63-Ub2 cleavage activity of LotB. These mutants lead us to identify additional hydrophobic interaction residues that were not described previously (Ma et al., 2020). All the mutations on LotC (E245R, M248R, Y119R, Y149R, E153R, Figure 4B, D, F) that result in the failure of K48-Ub2 cleavage are also newly identified from this study.

2) Interactors are not necessarily substrates – Whilst the title alludes to substrate ubiquitination, and the Introduction states "we…identified the specific host-substrates of LotB and LotC", the experiments are not directed to identify the substrates – i.e. proteins that may be less ubiquitinated in the presence of the Legionella DUBs. This would have required a different approach, for example isolating ubiquitylated proteins or peptides in the first place.The interactome datasets may of course identify interesting binding partners that may well include both regulators and substrates. However no orthologous validation experiments have been conducted to verify firstly the interaction and secondly assess the putative enzyme-substrate relationship. The figures focus on the comparison of interactors that are enriched in "mutant over empty" – presumably an empty plasmid – rather than drawing a comparison between wild-type and inactive mutants. Importantly, there is a priori no evidence that catalytically inactive DUBs would act as true substrate traps – they may misfold or mislocalise and for that reason associate non-specifically with a different subset of proteins. Thus in the absence of further work, the information value of these datasets is unclear.The introduction of some Legionella E3 ligases in the experimental set-up does alter the interactome data but this could be due to an overall change in the ubiquitin landscape upon the overexpression of those E3s. In other words, overexpression of any other E3 may also impinge on the interactome associated with the Legionella DUBs. This does not necessarily imply a coordination of activities in the host. The overall rationale of those experiments is not clear – the Legionella E3 ligases do not necessarily modify host proteins that are then targeted by Legionella DUBs.

We highly appreciate reviewer’s comments that interactors are not necessarily substrates, introduction of E3 ligase can cause overall change in the ubiquitin landscape and lack of evidence that catalytic mutant can trap ubiquitinated substrates. In some of our interactome datasets we could not find ubiquitin as highly enriched while comparing Mut over Wt DuB. Hence, we decided to show Mut over empty vector for all the interactome data while to tackle the real substrate identification issue, we developed another MS approach as described below.

As reviewer suggested, to identify substrates we developed new MS approaches where we enriched the ubiquitinated substrates under Legionella infected condition. Since reviewer concerned that the catalytically inactive mutant may not trap substrates, we checked the level of ubiquitinated proteins from both wild-type and mutants (Figure 6A). As shown, catalytically inactive LotB and LotC enriched more ubiquitinated proteins from Legionella infected cells (Figure 6B, C). To avoid another concern that this enrichment is the result of mis-fold or mis-localization of DUBs, we validated 6 substrates and performed the in vitro deubiquitination assay (Figure 6F-H). Together, we have now provided list of putative substrates for both LotB and LotC with orthogonal validation data.